# Novel Microfluidics Device for Rapid Antibiotics Susceptibility Screening

**Emil Grigorov [1,\*], Slavil Peykov [2,3] and Boris Kirov [1,2,4]**

1    Faculty of German Engineering Education and Industrial Management (FDIBA), Technical University of Sofia, 1756 Sofia, Bulgaria; boris.kirov@tu-sofia.bg
2    BioInfoTech Lab, Sofia Tech Park, 1784 Sofia, Bulgaria; spejkov@biofac.uni-sofia.bg
3    Department of Genetics, Faculty of Biology, Sofia University "St. Kliment Ohridski", 1164 Sofia, Bulgaria
4    Department Industrial Automation, Technical University of Sofia, 1756 Sofia, Bulgaria
*    Correspondence: emil.grigorov@fdiba.tu-sofia.bg

**Featured Application: Microfluidics device for rapid antibiotics susceptibility screening for bacterial strains in laboratory environment.**

**Abstract:** In recent years, excessive utilization of antibiotics has led to the emergence of antibiotic microbial resistance on a planetary scale. This recent phenomenon represents a serious threat to public health, as well as an enormous burden for healthcare systems' budgets worldwide. Novel, rapid and cheap methods for antibiotic susceptibility screening are urgently needed for this obstacle to be overcome. In this paper, we present a microfluidic device for on-chip antibiotic resistance testing, which allows for antibiotic microbial resistance detection within 6 hours. The design, fabrication and experimental utilization of the device are thoroughly described and analyzed, as well as possibilities for future automation of the whole process. The accessibility of such a device for all people, regardless of economic status, was of utmost importance for us during the development of the project.

**Keywords:** microfluidics; vacuum loading; diffusion modeling; antibiotic susceptibility; low cost

## 1. Introduction

Since the appearance of the first commercially available antibiotic in the late 1930s and their massive introduction into modern medicine in the 1940s, they have become an essential part of civilization as we know it. Indeed, their high efficiency and affordability made them a popular cure among individuals from all around the world. In the last few decades, however, the extensive use of antibiotics led to the emergence of microorganisms that are resistant to previously known efficient suppressing agents [1]. Consequently, the treatment of some infections became somewhat difficult, raising general morbidity and mortality and, as a result, increasing economic burden on healthcare systems. For example, between 2005 and 2015, almost one in every five infections in the EU was due to antibiotic-resistant bacteria. The resistance to second-line and third-line antibiotics increased by 20% and 35%, respectively, in 2010 compared to 2005 and is also expected to grow by more than 72% by 2030. Each year, antimicrobial resistance (AMR) is accountable for about 33,000 deaths and about 1.1 billion Euros in cost to the healthcare systems of European countries [2]. Similar tendencies were reported for the USA [1] and the Asia Pacific region [3].

In order to tackle the issue of AMR, it is mandatory to develop fast, cheap and reliable experimental methods for antibiotics susceptibility screening of microbial cells. There are several techniques typically utilized for the monitoring of antibiotic resistance. Most of the methods (broth dilution [4], disk diffusion [5], polymerase chain reaction (PCR)-based resistant gene detection [6]) are based on evaluation of bacterial growth on agar plates or molecular detection of resistant genes. The main drawbacks of the aforementioned

techniques are connected to long incubation times and poor quantification capabilities [7]. To address these issues, novel AMR methods have been designed and assessed, such as monitoring of bacterial biofilm evolution through optical and electrical measurements [8], matrix-assisted laser desorption/ionization time-of-flight (MALDI-TOF) [9], Raman spectroscopy [10], digital time-lapse microscopy [11] and microarrays [12]. Even though some of those techniques provide accurate antimicrobial resistance (AST) determinations in an ultra-rapid manner (2–3 h reported in [13] versus 8–10 h with automated conventional AST systems [14,15]), their complexity, high accompanying costs and the need for trained staff make them rather unaffordable tools for most laboratories worldwide. Furthermore, test result dissemination speeds remain limited. As indeed described in [16], less than 5% of European laboratories could validate and transmit the results of an AST to clinicians 24 h/day without delay. The even lower laboratory capacities for testing and reporting of results, combined with the inadequate access to basic health services, make it almost impossible for people living in developing countries to have access to such novel ultra-rapid technologies [17]. Additionally, the current health crisis provoked by the COVID-19 pandemic has demonstrated that supply chains are fragile and clinician specialists are insufficient even in developed countries. Therefore, it seems essential to focus research efforts as much as possible towards low-cost, stand-alone and specialist-independent technologies if we are to provide adequate care to as many people as possible without discrimination.

In recent years, microfluidics emerged as a dominant approach to solve a number of questions arising from the limited observability of biological processes at the single-cell level [18]. Concisely, microfluidic devices consist of interconnected fluidic channels and chambers having at least one of their dimensions smaller than a millimeter, i.e., micron-scale, hence "microfluidics". The most important aspect of the physics of those devices is the extremely low Reynolds number (fluid inertia vs. viscosity ratio), resulting in strictly laminar flow. The latter provides microfluidic devices with the unique property of absolute control over fluidic flows and reactants supply in different chip regions. Additionally, the decreased reagents volumes, usually in the micro/nanoliters range, can significantly reduce reaction times, energy consumption and overall cost for a certain process [19]. By providing well-defined microenvironments that mimic the natural habitats of bacteria, microfluidic devices have also enabled successful antibiotic susceptibility testing, on-chip cultivation, etc. [20]. Some of the many advantages of microfluidic devices utilized for antibiotic resistance screening include live monitoring of bacterial growth dynamics, rapid readout of resistance test results (usually within a few hours) and high potential for automation [20]. One example was reported in [21], where the susceptibility of *E. coli* to antibiotics was quantified on the basis of measurements of fluorescence intensity within 4 h. In [22], a microfluidic point-of-care device utilized for rapid antibiotic prescription assistance in cases of urinary tract infections caused by *E. coli* was presented. The growth rate of individual cells under the influence of antibiotics was determined based on automated phase-contrast microscopy at a single-cell level and subsequent image processing. The results were detectable within 30 min from sample loading to test readout. A detailed review of existing microfluidic devices for AST is described in [23].

The aforementioned devices indicate that the utilization of microfluidics can accelerate the AST of bacterial cells even further. The main drawbacks of those techniques remain the necessity of advanced detection devices (e.g., single-cell microscope) and/or the presence of expensive soft lithography machinery for microfluidic chip manufacturing [24]. In this paper, we present a novel microfluidic device for on-chip AST. We describe a novel cost- and time-efficient method for device fabrication without the need for a clean room environment. The distinguishing capacity of polydimethylsiloxane (PDMS) for gas permeability combined with dedicated vacuum channels was utilized in order to engineer a cultivation device that allows for reliable inoculation in a semi-automatic manner. This way, we were able to populate more than 100 independent cultivation chambers with *E. coli* cells and efficiently monitor their growth in the presence/absence of various antibiotic compounds.

## 2. Device Description

As mentioned before, our device utilizes the gas permeability of PDMS [25]. The main goal was to develop a microfluidic device in which the influence of antibiotics on the bacterial cell growth rate can be visually observed. The working principle of the designed microfluidic chip was inspired by [26], where the presented technique successfully captured mammalian cells in isolated regions, protecting the cells from potential flow-damaging effects. The design of the device is shown in Figure 1a. The chip consists of three channels: two cultivation channels (220 μm × 220 μm, 200 μm height), each having 69 individual culture chambers (300 μm × 300 μm, 200 μm), and one vacuum channel, running parallel to the column of chambers at a distance of 300 μm, as shown in Figure 1b. The cell traps are connected to the flow in the cultivation channels through narrow channels (80 μm × 150 μm, 200 μm), which ensures that mass transport mostly occurs through diffusion. The main idea of the chambers is to allow the control of cell loading at low velocities without the need for high external pressure. The application of a vacuum in the middle microchannel allows for air evacuation from the chambers, followed by chamber cell loading. This way, the chambers protect the cells from the shear forces in the main flow channel and, most importantly, serve as microincubators, in which cells could develop without any disruption. According to our knowledge, we are the first to report the vacuum loading of bacterial cells in a microfluidic device. In contrast to [26], we utilized a vacuum pump for the air evacuation (described in Section 4), allowing for a higher degree of automation in the process.

**(a)** **(b)**

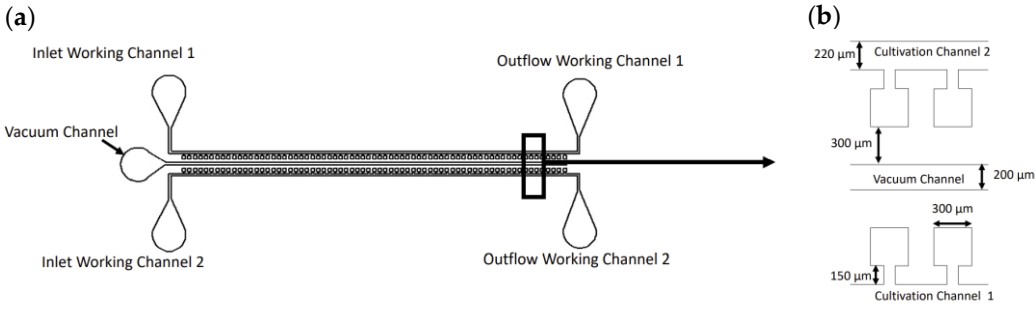

**Figure 1.** (**a**) Schematic of the utilized microfluidic device; (**b**) zoomed view on the cell chambers with their dimensions.

In order to choose the optimal length of the narrow channel, we made a finite volume analysis, comparing the mass transport in the culture chambers for four different lengths. The results are described in the section below. After cells are loaded in the channel's chambers through the vacuum, an antibiotic is introduced in Cultivation Channel 1 and nutrient-rich media in Cultivation Channel 2. This way, the effect of the antibiotic on the cell culture growth rate compared to pure medium could be inspected through direct observation.

## 3. Flow Modeling

As already discussed, in order to choose the optimal length of the narrow channel, connecting the chambers with the main flow of the cultivation channels, we carried out finite-volume-modeling of the experimental flow conditions, utilizing Ansys Fluent 2021R1. Since the fabrication of the channels was constrained due to the resolution of the utilized printer (described in the section below), the width of the narrow channel was fixed at 80 μm. The main question therefore remained: how long should the narrow channel be in order to obtain optimal flow conditions, and how long would it take for the fluid from the main channels to reach the inside of the traps?

Since the flow going inside the chambers is mostly driven by diffusion phenomena, we utilized the Species Transport Model [27], which predicts the local mass fraction of each species through the solution of a convection-diffusion equation. The molecular

concentration differences of the utilized working fluids (Lysogeny broth (LB) with or without Chloramphenicol, described in the sections below) are negligibly small compared to water; hence, pure water was used for the simulations. The data for the self-diffusion coefficient of water were taken from [28] at D = 2.29 $\mu m^2/ms$. The results for the four narrow channel lengths are shown in Figure 2.

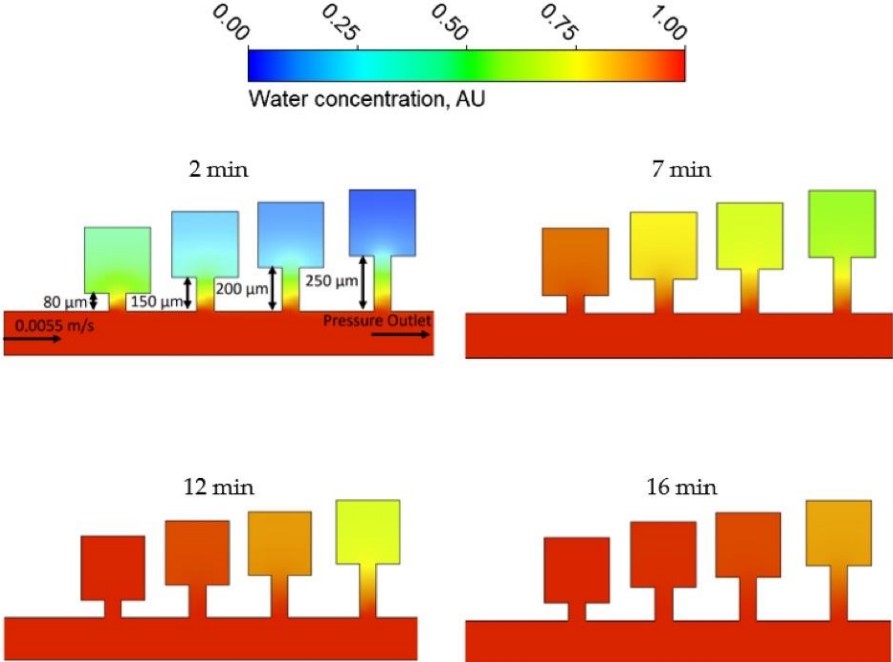

**Figure 2.** Results of the flow modeling simulations, showing the water concentration in arbitrary units (AU) in the chambers for four different times.

From Figure 2, it is clear that for all four of the analyzed geometries, the fluid penetrating the chambers from the cultivation channels manages to infuse the whole trap, leaving no zones without the supply of fresh medium. However, due to the constraints of fabrication process precision, we were forced to utilize the second smallest chamber, which was expected to provide capacity for relatively fast bacterial growth. Last but not least, the simulation results clearly demonstrate that the time required for equilibration between the concentrations of small molecules, such as Chloramphenicol (CA), in our preferred chip geometry and the supply channel (16 min) is much shorter than the waiting period before the beginning of the measured observations (4 h).

## 4. Device Fabrication

Traditional fabrication methods for microfluidic devices are usually based on soft lithography and require a number of specialized instruments, which can only be used in clean rooms, and even subcontracting some activities, such as film mask fabrication. The latter could prolong the period between device design and fabrication up to a couple of weeks. In contrast, our microfluidic device was fabricated using a mold produced by a resin LCD-based stereolithographic type 3D printer (Photon Mono, Shenzhen Anycubic Technology Co., China), with a printing XY-resolution of 50 $\mu m$ (2560 × 1620). In practice, however, the achieved resolution was about 65–70 $\mu m$. The mold was made with the help of a computer-aided design (CAD) software (SolidWorks 2020, Dassault Systems, France), and the resulting data were then transferred to the printer's software. The mold was printed with a UV-sensitive resin, washed with isopropanol and cured for 8 min with a 395 nm UV light. In the next step, a mixed PDMS pre-polymer was poured into the printed mold and cured in an oven at 60 °C for about 1 h. After peeling it off from the resin mold, the hardened PDMS and a microscope cover slip were treated with oxygen plasma

produced by plasma cleaner (Diener electronic PCCE) and attached to each other by baking at 60 °C for 12 h. The whole procedure is represented schematically in Figure 3. Overall, our original cost- and time-efficient fabrication process required only a plasma cleaner and a low-cost 3D printer, and less than 24h between the finalization of the CAD model and the loading of the bacterial cultures.

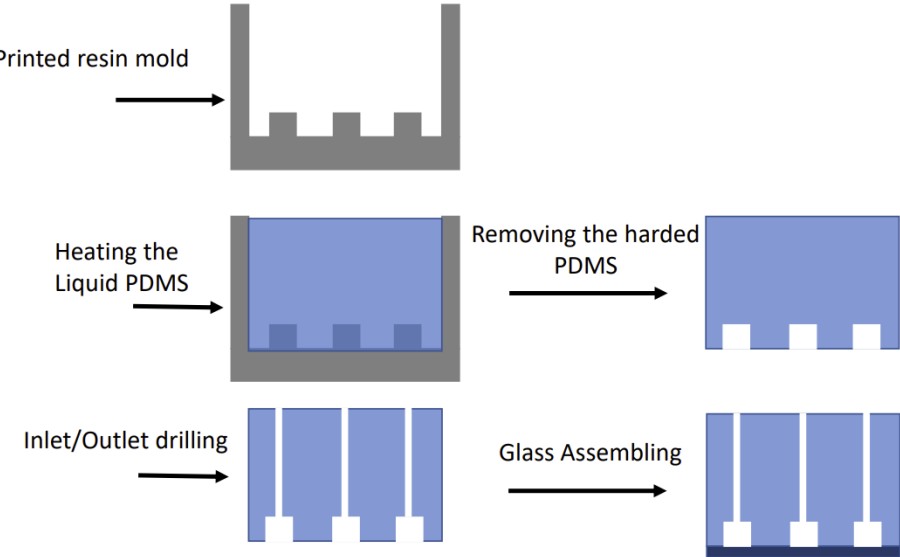

**Figure 3.** Schematic of the device fabrication steps.

## 5. Microfluidics Operation and Antibiotic Susceptibility Testing Setup

*Escherichia coli* KRX cells have been utilized in all experiments described here. This strain is a commercially available derivate of *E. coli* K-12, created by Promega (catalog no. L3002 (Madison, WI, USA)). KRX cells are intensively used as a suitable host for various molecular cloning procedures that include high-yield protein expression with tight regulation. The cells were transformed with a high copy number pSB1C3 vector that confers resistance to CA. They were cultured in a Lysogeny broth (LB) medium and left overnight in a shaker incubator at 37 °C. The culture was then refreshed by dilution 500 times and cultivation for 4 h before being manually loaded into the two cultivation channels via a 3 mL syringe, as shown in Figure 4a. To prepare the vacuum, an 8 mm ID hose was connected to a vacuum pump (Vacuubrand ME 1C, Vacuumbrand GmbH, Germany) on one side and via an adapter interface to a 0.6 mm ID microfluidic tubing on the other side. With a 0.6 mm OD PDMS coupler, the connection was finally inserted into the port of the vacuum channel (Figure 4b). The pump was then turned on, and the chamber filling was visually monitored. The filling of the chambers lasted approx. 15 min. Figure 5 shows different time frames from the process. Once the traps were completely filled with cells/fluids, the pump was turned off, and the vacuum connections were carefully disconnected from the device.

Since we placed the device on a 37 °C heated plate, in order to achieve the optimal conditions for the bacteria growth, an increased rate of fluid evaporation inside the traps had to be expected due to high surface-area-to-volume ratios in the system. To overcome this problem, a constant flow of media was used to refresh the culture volume. For our device, we found that a flow ratio of 0.8 mL/h is high enough to ensure the permanent presence of fluid even after a long duration of experiments. Two syringe pumps with 60 mL syringes were used to obtain the mentioned flow rate value inside the two cultivation channels, as shown in Figure 4c. The first one was filled with CA (34 µg/mL) + Lysogeny broth (LB). The second cultivation channel was utilized to mimic the natural habitats of the bacterial cells and therefore filled only with LB as a highly nutrient-rich medium.

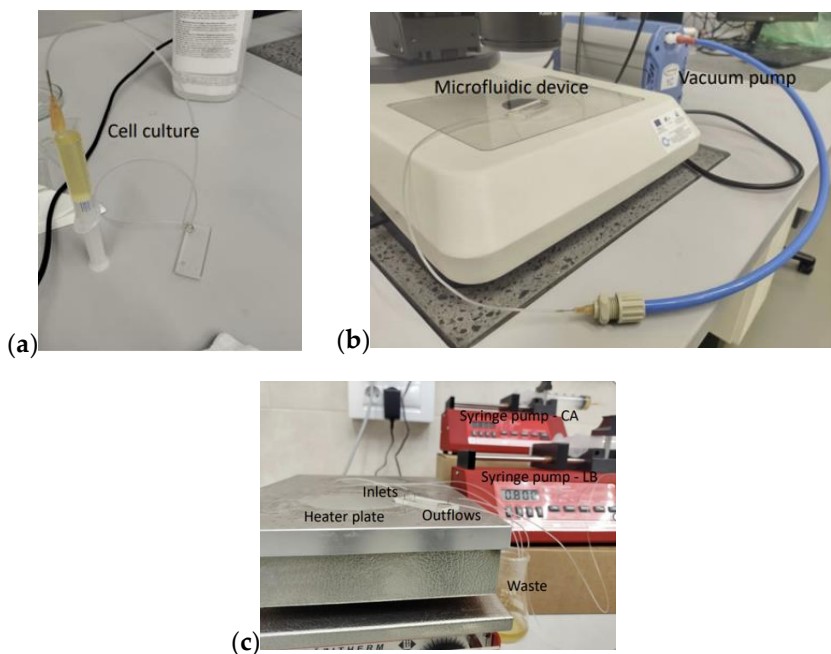

**Figure 4.** Experimental setup: (**a**) Cell loading in the device; (**b**) polydimethylsiloxane (PDMS) degassing through a vacuum pump; (**c**) two syringe pumps obtained the Chloramphenicol (CA) and Luria broth (LB) in the two cultivation channels. The device was placed on a heated plate (37 °C).

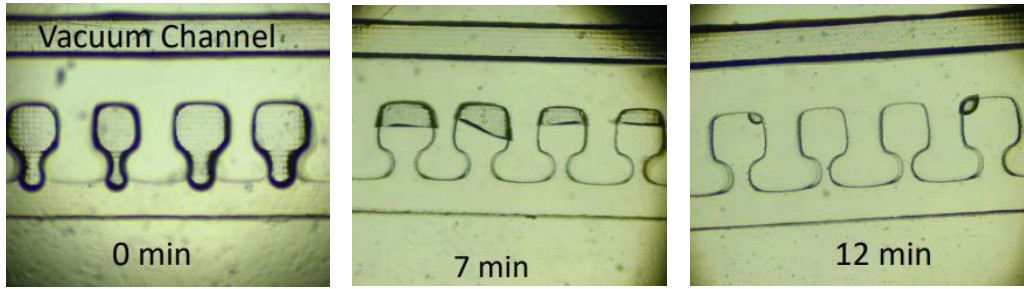

**Figure 5.** Different time frames of the chamber filling process. The complete air evacuation lasted approx. 15 min.

## 6. Results and Discussion

Our objective in this study was to design a microfluidic chip, which enables the direct observation of the growth of bacterial strains with antibiotic resistance. To test this, we monitored the cell growth inside the device growth chambers throughout a 12 h experiment. Figure 6 shows the bacterial development in three individual traps at three different time frames after the beginning of the experiment (4 h, 8 h and 12 h).

A clear tendency in the growth rate of the cells is visible in the chambers where pure LB is the working fluid. In order to describe this process quantitatively, we analyzed the captured pictures of cell chambers through the experiment. The average intensity of the pixels served as an estimation of cell growth. The data for each of the observed chambers were normalized towards the light intensity measured in the chambers positioned right across the supply channels on the same microfluidic device in the same time instances within the same experiment. The objective of this normalization was to take into account instrumental light drift and/or possible growth of cells in the chambers with an antibiotic. Figure 7 represents cell development as a function of time and the corresponding error bars, indicating one standard deviation of uncertainty. Every data point represents the average intensity of three captured images at the corresponding time. The dashed line depicts the measured raw light intensities of the control chambers. A clear change in the cell growth speed is visible somewhere at the sixth hour. Compared to a traditional phoenix system,

our device shortens the needed result time by 4–6 h [14,15]. Even though the results of other microfluidic chips look more promising (1–4 h), as described in the introduction section, their utilization is mostly characterized by the need for an expensive and complex detection sensor at the single-cell level, which makes them unsuitable as an affordable point-of-care device.

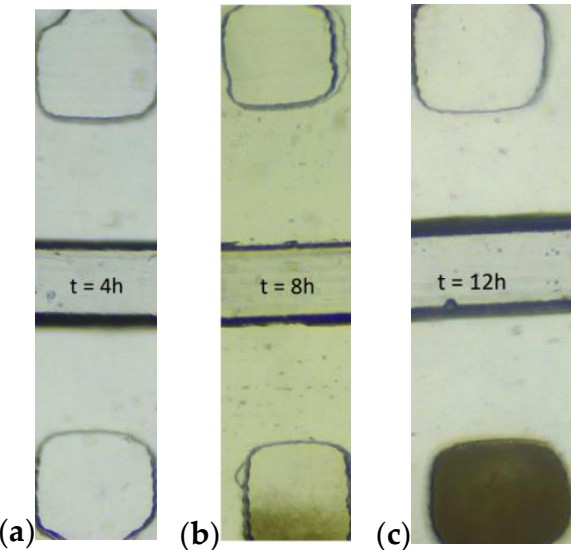

**Figure 6.** Bacteria growth in the chambers at different time frames, 4 h (**a**), 8 h (**b**) and 12 h (**c**) after the start of the experiment. LB + CA in the channel above, pure LB in the channel below.

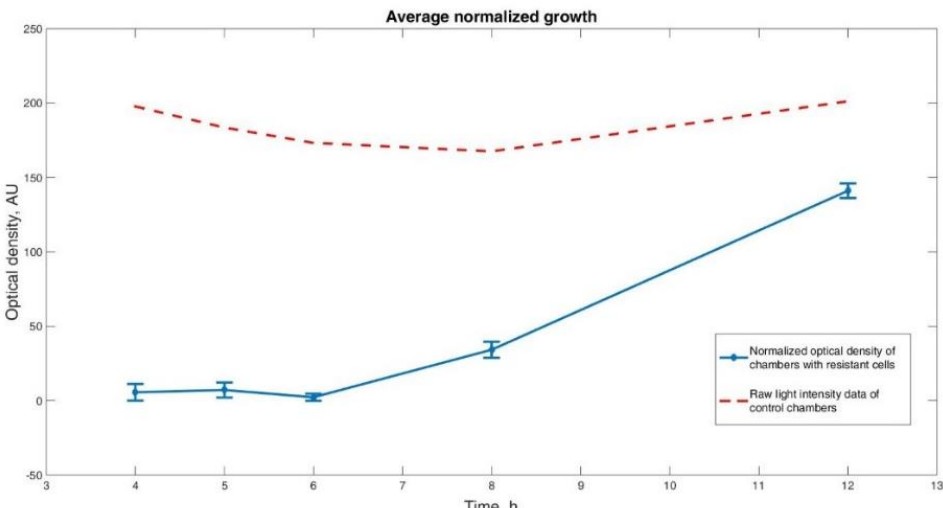

**Figure 7.** Normalized cell growth. Three sets of opposing growth chambers were analyzed for each time point. Optical density of the chambers without added antibiotic normalized against the chambers with antibiotic was taken as a measurement of cell growth. It is visible that the growth of cells in the chambers with pure LB at the given experimental conditions becomes observable after the sixth hour.

Figure 8 is a direct observation (smartphone picture without zoom) of the microfluidic device after the complete 12 h experiment was conducted, showing a clearly visible difference in the light absorption between the chambers with CA + LB from the chambers with pure LB.

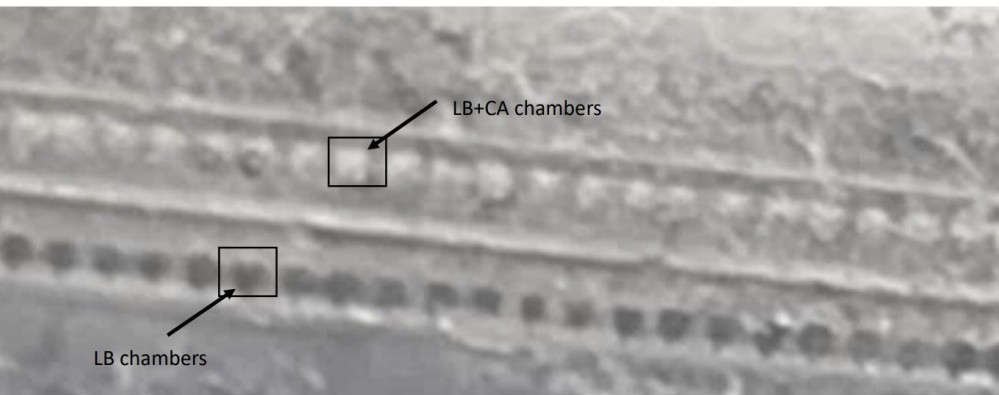

**Figure 8.** Naked-eye view of the microfluidic device after the experimental part was conducted.

## 7. Conclusions

In this paper, we presented the engineering of a novel microfluidic device for rapid screening for bacterial antibiotic susceptibility. We utilized an innovative approach for growth chamber inoculation with vacuum, which allowed for the application of a simple single-layer design. The latter was successfully fabricated in a low-cost setup based on resin 3D printing. Our experimental results show that the difference between the microbial growth within the chambers fed with pure medium and antibiotic-enriched medium becomes clearly distinguishable after the sixth hour of cultivation. The latter was also confirmed by directly analyzing the pixel intensity of the images captured by the microscope camera. Consequently, our method is prone to automation not only of the inoculation but also of the cell AMR screening process. Additionally, the time frame for results observation is somewhere around the average of previous reports of such devices.

Overall, the presented device and the conducted experiments are a good example of a microfluidic system that could be utilized as an antibiotic resistance detection system, which exemplifies a reliable, reproducible and, most importantly, cost-effective experimental method for microbial cell screening.

The vacuum-loading approach of the growth chambers proved to be quite successful, efficient, and seemingly independent of the chambers' connections (shape and size) to the main feeding channel. The latter represents a significant paradigm shift in microfluidic device design with new horizons yet to be discovered and appreciated.

Finally, in our endeavor to optimize the design, we have stripped the proposed device from all unessential functions and requirements for additional instruments and specialized knowledge. After all, in our setup, it is completely possible to exchange the vacuum pump with manual suction with a syringe, the driving force of the syringe pump with gravitational flow (i.e., attach the syringe to the wall at sufficient height with adhesive tape), the plate heater with any source of body temperature heat (including a human body itself) and, finally, the cultivation medium with a simple urine sample.

**Author Contributions:** Conceptualization, B.K. and E.G., methodology, S.P.; writing—original draft. All authors have read and agreed to the published version of the manuscript.

**Funding:** This research was funded by America for Bulgaria Foundation, grant num.

**Institutional Review Board Statement:** Not applicable.

**Informed Consent Statement:** The totality of the experimental and theoretical work for this research was conducted on the premises of the BioInfotech lab of Sofia Tech Park (Sofia, Bulgaria).

**Data Availability Statement:** Not applicable.

**Conflicts of Interest:** The authors declare no conflict of interest.

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
