# Peer review of "Novel Microfluidics Device for Rapid Antibiotics Susceptibility Screening"

_applsci, doi:10.3390/app12042198_

Round 1

Reviewer 1 Report

Major concerns:

The authors highlight standard methods of antimicrobial susceptibility testing, but largely overlook the fact that there are other methods that have been suggested to look at antimicrobial susceptibility in shorter time frames.  Given the focus of this report – reference to other rapid testing strategies is essential and should also include a discussion on why a microfluidics based approach would be preferrable (if it is more biologically relevant discuss that in more detail, how does it mimic infection?). For example this is expected to be the case if testing susceptibility of microbial biofilms in dynamic environments. 

The purpose of the study was “to design a microfluidic chip, which enables the direct observation of growth of bacterial strains with antibiotic resistance.”   I agree that the design part is there, but part of your design is to ensure that it can be used to grow bacteria and determine antimicrobial susceptibility. It is therefore rather surprising that no antimicrobial susceptibility tests were performed, and only a single strain of E. coli at a single concentration of chloramphenicol was used.  An experiment performed to demonstrate the differential growth of a strain that is resistant and one that is susceptible to the same drug would be warranted.  The article gives no indication on the potential utility of this product without at least preliminary assessments of accuracy and precision.

This article left me with far more questions than answers.  It’s critical that the purpose of the chambers be explained in more detail and it must be clear if they are intended to be used as replicates of the antibiotic and bacteria concentration.   If the microfluidic device is flowing to refresh the culture volume (line 140), but it also appears to have waste leaving the flowcell how can you ensure that concentration of the antimicrobial is constant in all chambers throughout the length of the experiment (the answer can be implied if you look through the details, but I think it is critical that such things are clearly communicated in you descriptions).  To test the MIC of one strain would you therefore need at least 6 flow cells to get 12 antibiotic concentrations that could be tested by standard broth microdilution – and potentially read within 6 hours using a plate reader as well?  I strong urge the authors to review https://doi.org/10.1016/j.cmi.2019.04.025

Minor Concerns: 

Line 25 – remove “to the”

Line 32-33 missing citation for 2005-2015 data.

Line 41 – remove “(“

Lin 56 – lowercase p for polydimethylsiloxane

Line 60 – italicize Escherichia coli

Line 76 – It is very unclear why the authors suggest here that the chambers will provide a hospitable place for bacteria to grow “without any disruption” when the modeling clearly suggests fluid exchange within 16 minutes (Figure 2).  I

Line 86 – suggest optimal instead of “correct”’

Line  95-missing reference for species transport model

Line 100 – missing reference

Lines 102-106. The authors discuss here that different geometries of the chambers were tested, however figure 2 appears to show a time course of a single chamber size.  Is there a figure missing, or is this figuring showing various sized chambers as well as a time course?  If so it is perplexing why different sized chambers would not be compared after the same length of time.

Line 144 – what concentration of chloramphenicol was used (or what weight for the flow cell?)

Figure 1 – strongly suggest to rotate part b to correspond with the same orientation at part a.  Or please note that this is rotated 90° clockwise/counterclockwise.

Figure 2 – legend is cutoff.

Figure 3 – suggest to move text within the figure to space next to each image for clarity and ease for the reader.

Figure 7 – it is unclear if the data represents 3 experimental chambers compared to 3 control chambers on the same flow cell, or if 3 independent flow cells were used for data collect.  Both are critical for establishing this in order to demonstrate both intra- and inter-assay precision.  The y axis title does not reflect normalized cell growth.  One would assume that normalized would be experimental AU / initial AU.  Was only the control plotted?  What about the data of the CA trial?  I would like to see that plotted as well even if it’s a flat line.  Normalized cell growth should be described more in the legend as well. 

Experiments were done with LB, while the vast majority of antimicrobial susceptibility testing is performed with Mueller Hinton II broth - I would strongly urge the authors to use MHB in future assessments of this flowcell. 

Author Response

Major concerns:

The authors highlight standard methods of antimicrobial susceptibility testing, but largely overlook the fact that there are other methods that have been suggested to look at antimicrobial susceptibility in shorter time frames.  Given the focus of this report – reference to other rapid testing strategies is essential and should also include a discussion on why a microfluidics based approach would be preferrable (if it is more biologically relevant discuss that in more detail, how does it mimic infection?). For example this is expected to be the case if testing susceptibility of microbial biofilms in dynamic environments. 

 Authors response: Thank you for your valuable suggestion. We have added more detailed overview regarding more modern approaches for the AMR phenomenon (lines 47-65). Moreover, we added information regarding the advantages of the microfluidic approach and emphasized the novelty of our device with respect to already existing microfluidic methods (lines 66-89). In general a microfluidics based approach shortens the reaction times by couple of factors, which is essential from engineering point of view when dealing with antibiotic susceptibility screening. Our main reasons to adopt microfluidics was the possibility for low-cost fabrication and integration, and automation of many of the processes which require specialized labor and expensive instrumentation. The device is aimed to be widely used, no matter the wealth of the nation or the individual, which was also added as a phrase in the abstract.

The purpose of the study was “to design a microfluidic chip, which enables the direct observation of growth of bacterial strains with antibiotic resistance.”   I agree that the design part is there, but part of your design is to ensure that it can be used to grow bacteria and determine antimicrobial susceptibility. It is therefore rather surprising that no antimicrobial susceptibility tests were performed, and only a single strain of E. coli at a single concentration of chloramphenicol was used.  An experiment performed to demonstrate the differential growth of a strain that is resistant and one that is susceptible to the same drug would be warranted.  The article gives no indication on the potential utility of this product without at least preliminary assessments of accuracy and precision.

Authors response: We absolutely agree with the above remarks. The experiments for device validation also including AST of real clinical isolates from patients with urinary tract infections are ongoing. We also engineered a device capable of assessing 5 different antibiotic compounds, but we need more time for the experiments. All those results shall be reported in another paper. Here we aimed towards the proof of principle of the device functional and fabrication efficiency. 

This article left me with far more questions than answers.  It’s critical that the purpose of the chambers be explained in more detail and it must be clear if they are intended to be used as replicates of the antibiotic and bacteria concentration.   If the microfluidic device is flowing to refresh the culture volume (line 140), but it also appears to have waste leaving the flowcell how can you ensure that concentration of the antimicrobial is constant in all chambers throughout the length of the experiment (the answer can be implied if you look through the details, but I think it is critical that such things are clearly communicated in your descriptions). 

Authors response: Thank you for the remark. We added few explicit explanatory lines regarding why we assume that the antibiotics concentration is constant within the context of constant supply (Lines 150-154).

To test the MIC of one strain would you therefore need at least 6 flow cells to get 12 antibiotic concentrations that could be tested by standard broth microdilution – and potentially read within 6 hours using a plate reader as well?  I strong urge the authors to review https://doi.org/10.1016/j.cmi.2019.04.025

Authors response: Once again, we agree with your remark. However, it is worth to mention that the intention of our device was never to determine MIC’s according to the traditional approach, but to provide means for rapid assessment if the given antibiotic in its therapeutic concentration is suitable for a patient at the initial stage of a urinary tract infection.

Minor Concerns: 

Line 25 – remove “to the”

 Author response: Thank you for pointing this out, we made a revision and removed the suggested phrase.

Line 32-33 missing citation for 2005-2015 data.

 Author response: Thank you for pointing this out, we included also information for the year 2010 (Line 35).

Line 41 – remove “(“

 Author response: Thank you for pointing this out, we’ve made a revision and fixed the typos we found, including the suggested.

Lin 56 – lowercase p for polydimethylsiloxane

  Author response: Thank you for pointing this out, we changed the uppercase to lowercase (Line 95).

Line 60 – italicize Escherichia coli

 Author response: Thank you for pointing this out, we made the needed corrections (Line 98).

Line 76 – It is very unclear why the authors suggest here that the chambers will provide a hospitable place for bacteria to grow “without any disruption” when the modeling clearly suggests fluid exchange within 16 minutes (Figure 2). 

 Author response: Thank you for the comment, the main idea of the chambers was to isolate the cells in regions which are dominated by diffusion transport and where the shear effects of the main channel are negligible. The exchange of fluid withing 16 minutes is due to diffusion, it occurs on molecular level and does not disrupt the cells inside the chambers by any means. The only purpose of this exchange is to supply the traps with fresh medium. In short, this type of device is a microscopic version of a chemostat.

Line 86 – suggest optimal instead of “correct”’

 Author response: Thank you for pointing this out, we made a revision and corrected it (Line 130).

Line  95-missing reference for species transport model

Author response: Thank you for pointing this out, we added the reference (Ref 28).

Line 100 – missing reference

 Authors response: Thank you for pointing this out, we added the missing reference (Ref 29).

Lines 102-106. The authors discuss here that different geometries of the chambers were tested, however figure 2 appears to show a time course of a single chamber size.  Is there a figure missing, or is this figuring showing various sized chambers as well as a time course?  If so it is perplexing why different sized chambers would not be compared after the same length of time.

 Authors response: Thank you for the comment, Figure 2 compares single chamber size with 4 different narrow channel lengths (the channel connecting the traps with the main channel) at different time steps and labels were rearranged for clarification. The main question was if for some reason there are ‘’dead zones’’ inside the traps without the supply of fresh medium. An additional parameter study regarding the length of the chambers was not conducted. 

Line 144 – what concentration of chloramphenicol was used (or what weight for the flow cell?)

Autors response: Thank you for the comment. The concentration we used was 34 ug/ml (Line 200), which is typical for selection of recombinant E. coli cells with high copy number plasmid vectors with CamR genes like the one we utilized (Lines 179-180)

Figure 1 – strongly suggest to rotate part b to correspond with the same orientation at part a.  Or please note that this is rotated 90° clockwise/counterclockwise.

 Authors response: Thank you for pointing this out, we rotated the mentioned Figure.

Figure 2 – legend is cutoff.

 Authors response: Thank you for pointing this out, we fixed the legend.

Figure 3 – suggest to move text within the figure to space next to each image for clarity and ease for the reader.

 Authors response: Thank you for pointing this out, we moved the text in the mentioned Figure as suggested.

Figure 7 – it is unclear if the data represents 3 experimental chambers compared to 3 control chambers on the same flow cell, or if 3 independent flow cells were used for data collect.  Both are critical for establishing this in order to demonstrate both intra- and inter-assay precision.  The y axis title does not reflect normalized cell growth.  One would assume that normalized would be experimental AU / initial AU.  Was only the control plotted?  What about the data of the CA trial?  I would like to see that plotted as well even if it’s a flat line.  Normalized cell growth should be described more in the legend as well. 

Authors response: Thank you indeed for this remark. The data for each of the observed chambers was normalized towards the light intensity measured in the chambers positioned right across the supply channels on the same microfluidic device in the same time instances within the same experiment. The objective of this normalization was to take into account instrumental light drift and/or possible growth of cells in the chambers with antibiotic. Figure 7 represents the cell development as a function of time and the corresponding error bars, indicating one standard deviation of uncertainty. Every data point represents the average intensity of 3 captured images at the corresponding time. The dashed line depicts the measured raw light intensities of the control chambers. All of the above has been added to the main text of the paper, as well as summary in the figure caption.(Lines 212-220).

Experiments were done with LB, while the vast majority of antimicrobial susceptibility testing is performed with Mueller Hinton II broth - I would strongly urge the authors to use MHB in future assessments of this flowcell. 

Authors response: We completely agree with the remark. We used LB because it is typical medium for maintenance and cultivation of recombinant E. coli strains, but all planned future work will be performed in MH broth.

Reviewer 2 Report

The Authors propose a microfluidics device for antibiotics susceptibility testing. The device has been designed by using Ansys Software. Experimental results confirm the suitability of the proposed device for the target application. Although the results are interesting, some contents should be mandatory clarified for the manuscript publication suggestion.

Here, my comments to the manuscript:

  • In the last years, a great research effort has been spent to contain the AMR phenomenon. In the Introduction Section, the Authors should cite the main technologies developed to contrast the phenomenon (see as example “Monitoring of individual bacteria using electro-photonic traps” Biomedical optics express, 10(7), 3463-3471, 2019; “Gram-type differentiation of bacteria with 2D hollow photonic crystal cavities,” Applied Physics Letters, 113(11), 111101, 2018; “Novel micro-nano optoelectronic biosensor for label-free real-time biofilm monitoring”, Biosensors, 11(10), 361,2021; “Detecting phenotypically resistant Mycobacterium tuberculosis using wavelength modulated Raman spectroscopy,” Antibiotic Resistance Protocols., 2018; “Rapid bacterial detection with an interdigitated array electrode by electrochemical impedance spectroscopy,” Electrochimica Acta, 82, 126-131, 2012). The aforementioned technologies are just few examples of a wide literature about device to contain AMR phenomenon. An overview of the technologies could help the reader to rate the performance of the proposed device.
  • About the novelty of the proposed device, the Authors report that the device is inspired by Ref. [11] of the manuscript. The Authors should highlight the novelty with respect to Ref. [11] and other similar devices reported in literature.
  • In the Section 3, the design of the channels should be improved. In particular, a parametric study about the length should be reported in order to rate the impact of the channel length on the device performance.
  • The style of the manuscript is debatable. In particular, I suggest to the Authors to avoid any questions in the manuscript. Furthermore, all figures should be improved, deleting the table contours and straighten up some subfigures.

Author Response

Reviewer 2

The Authors propose a microfluidics device for antibiotics susceptibility testing. The device has been designed by using Ansys Software. Experimental results confirm the suitability of the proposed device for the target application. Although the results are interesting, some contents should be mandatory clarified for the manuscript publication suggestion.

Here, my comments to the manuscript:

  1. In the last years, a great research effort has been spent to contain the AMR phenomenon. In the Introduction Section, the Authors should cite the main technologies developed to contrast the phenomenon (see as example “Monitoring of individual bacteria using electro-photonic traps” Biomedical optics express, 10(7), 3463-3471, 2019; “Gram-type differentiation of bacteria with 2D hollow photonic crystal cavities,” Applied Physics Letters, 113(11), 111101, 2018; “Novel micro-nano optoelectronic biosensor for label-free real-time biofilm monitoring”, Biosensors, 11(10), 361,2021; “Detecting phenotypically resistant Mycobacterium tuberculosis using wavelength modulated Raman spectroscopy,” Antibiotic Resistance Protocols., 2018; “Rapid bacterial detection with an interdigitated array electrode by electrochemical impedance spectroscopy,” Electrochimica Acta, 82, 126-131, 2012). The aforementioned technologies are just few examples of a wide literature about device to contain AMR phenomenon. An overview of the technologies could help the reader to rate the performance of the proposed device.

Authors’ response: Thank you for your valuable suggestion. We have added more detailed overview regarding the mentioned articles and few other techniques for the AMR phenomenon (lines 45-64). Moreover, we added information regarding the advantages of the microfluidic approach and emphasized the novelty of our device with respect to already existing microfluidic methods (lines 69-89).

  1. About the novelty of the proposed device, the Authors report that the device is inspired by Ref. [11] of the manuscript. The Authors should highlight the novelty with respect to Ref. [11] and other similar devices reported in literature.

Author response: Thank you for pointing this out, we made a revision and added a short description regarding the novelty with respect to Ref [11] (now Ref [27]) on Lines 107-110. In brief, we are the first to apply this type of loading for bacterial cells and we utilized vacuum pump for reliable vacuum generation. The latter also allows for further automation of the process within the framework of the engineering of a future electronic device (Lines 118-121).

  1. In the Section 3, the design of the channels should be improved. In particular, a parametric study about the length should be reported in order to rate the impact of the channel length on the device performance.

Author response: Thank you for this comment. Due to the big mesh size (ca 300 000 mesh elements) utilized in Section 3 and the computational resources of our lab, it is not possible for us to conduct such a parametric study within the 7 days provided for author’s response. We realize the importance of such study and we shall definitely try to include it in our future work.

  1. The style of the manuscript is debatable. In particular, I suggest to the Authors to avoid any questions in the manuscript. Furthermore, all figures should be improved, deleting the table contours and straighten up some subfigures.

Author response: Thank you for pointing this out, we made a revision and deleted any table contours and questions.

Reviewer 3 Report

The authors present an interesting paper on the developement of a Microfluidics Device for Rapid Antibiotics Susceptibility Screening. Despite the significance of content, the paper needs to be improved before publication.

In particular, the introduction is focused on the emergence of antibiotic-resistant bacteria. However, more attention should be paid on the microfluidic devices itselves. The authors cite them (line 46) without explain what a microlfuidic device is and without any reference. More details on the previus studies shoud be also provided: lines 46-54 are too generic and provide few useful details.

Moreover, It is not clear the novelty respect the state of the art?: the fabrication tehcnology? the materials? the geometry?  the processing time? Please, a comparison with other extisting devices for the same application must be provided.

Line 67-70: add a space between number and units.

Line 164: what do you mena for "no optics"?

The title "experimental setup" is misleading ( experimental setup for do what?) and should be changed with a  less generic title.

Line 174: Is this results promising compared to previous studies? What is usually the average time?

Author Response

Reviewer 3

The authors present an interesting paper on the development of a Microfluidics Device for Rapid Antibiotics Susceptibility Screening. Despite the significance of content, the paper needs to be improved before publication.

In particular, the introduction is focused on the emergence of antibiotic-resistant bacteria. However, more attention should be paid on the microfluidic devices itselves. The authors cite them (line 46) without explain what a microlfuidic device is and without any reference. More details on the previus studies shoud be also provided: lines 46-54 are too generic and provide few useful details.

Authors response: Thank you for your valuable suggestion. We have added more information regarding the advantages of the microfluidic approach and added a short overview of other existing methods utilized in the microfluidics for AMR (lines 69-89). A reference for the definition of microfluidics was added [Ref 19] (Line 66-75).

Moreover, It is not clear the novelty respect the state of the art?: the fabrication technology? the materials? the geometry?  the processing time? Please, a comparison with other extisting devices for the same application must be provided.

Author response: Thank you for pointing this out, we added a short description regarding the novelty with respect to Ref [11] (now Ref [27]) on Lines 117-121. The adding of a vacuum pump made the work with our device somewhat more automated, compared to [27]. Regarding the novelty of our device with respect to already existing microfluidic methods, we made a brief description in the introduction (Lines 89-95).

Line 67-70: add a space between number and units.

Author response: Thank you for pointing this out, we made a revision and added spaces between numbers and units now Lines 107-109.

Line 164: what do you mean for "no optics"?

Author response: Thank you for the comment, we changed ‘’no optics’’ to ‘’smartphone picture’’ (Line 227). The initial idea of this picture was to show that even without the need of a microscope setup the results can be seen with the naked eye.

The title "experimental setup" is misleading (experimental setup for do what?) and should be changed with a  less generic title.

Author response: Thank you for pointing this out, we changed the title of section 5 to ‘’Microfluidics Operation and Antibiotic Susceptibility Testing Setup ’’

Line 174: Is this result promising compared to previous studies? What is usually the average time?

Author response: Thank you for this comment, we compared our device and the evaluated times with conventional methods where approx. 10-12 h are needed for a successful antibiotic resistance detection (ref 15 and 16). Even though the results of other microfluidic chips can be more promising (1-4 h), compared to 6 h, they are mostly characterized by the need of expensive and complex detection sensors/devices at the single-cell level, which makes them more expensive for most laboratories worldwide and not suitable for point-of-care diagnostics.

Round 2

Reviewer 1 Report

Dear authors,

Thank you for your clarification of the scope of this current work - I do feel the wording in lines 233-234 suggests a level of testing not yet completed, however the additional emphasis on technology that is more broadly accessible reorients the reader and I commend the authors on this goal to provide rapid technologies that are accessible with minimal additional equipment & expertise.  This was a very valuable addition to the previous version.

Overall, I appreciate the responses and changes made to the manuscript, which has been much improved and I look forward to seeing the validation results of susceptibility measures using your technology in the future. 

Reviewer 2 Report

The Authors have replied to the Reviewers’ comments with satisfactory responses. The new version of the manuscript has been modified following the Reviewers’ suggestions. Therefore, I endorse the manuscript.

Reviewer 3 Report

The authors improved the paper following the suggestions, stressing the novelty with respect to the state of the art.